# Social Capital and the Improvement in Functional Ability among Older People in Japan: A Multilevel Survival Analysis Using JAGES Data

**DOI:** 10.3390/ijerph16081310

**Published:** 2019-04-12

**Authors:** Airi Amemiya, Junko Saito, Masashige Saito, Daisuke Takagi, Maho Haseda, Yukako Tani, Katsunori Kondo, Naoki Kondo

**Affiliations:** 1Department of Health Education and Health Sociology, Graduate School of Medicine, The University of Tokyo, 7-3-1, Hongo, Bunkyo-ku, Tokyo 113-0033, Japan; amemiya@m.u-tokyo.ac.jp (A.A.); j.junkosaito@gmail.com (J.S.); hasedam@gmail.com (M.H.); 2Faculty of Social Welfare, Nihon Fukushi University, Okuda, Miahamacyo, Chitagun, Aichi 470-3295, Japan; masa-s@n-fukushi.ac.jp; 3Department of Health and Social Behavior, Graduate School of Medicine, The University of Tokyo, 7-3-1, Hongo, Bunkyo-ku, Tokyo 113-0033, Japan; dtakagi-utokyo@umin.ac.jp; 4Department of Global Health Promotion, Tokyo Medical and Dental University (TMDU), 1-5-45, Yushima, Bunkyo-ku, Tokyo 113-8510, Japan; tani.hlth@tmd.ac.jp; 5Department of Social Preventive Medical Sciences, Center for Preventive Medical Sciences, Chiba University, 1-8-1, Inohana, Chuo-ku, Chiba 360-0856, Japan; kkondo@chiba-u.jp; 6Department of Gerontological Evaluation, Center for Gerontology and Social Science, National Center for Geriatrics and Gerontology, 7-430, Moriokacho, Obu-shi, Aichi 474-8511, Japan

**Keywords:** functional disability, Japan, multilevel analysis, older people, social capital

## Abstract

We investigated the contextual effects of community social capital on functional ability among older people with functional disability in Japan, and the cross-level interaction effects between community social capital and individual psychosocial characteristics. We used data from the Japan Gerontological Evaluation Study for 1936 men and 2207 women nested within 320 communities and followed for 46 months. We used objective data for functional ability trajectories derived from the national long-term care-insurance system, and a validated measure of health-related community social capital comprising three components: civic participation, social cohesion, and reciprocity. A multilevel survival analysis with a community-level random intercept showed that in communities with high civic participation, women who actively participated in any community group showed greater functional ability improvement than did women who did not participate (*p*_interaction_ = 0.05). In communities with high social cohesion, older men who perceived that their communities’ social cohesion was high showed greater functional ability improvement than men who perceived it to be low (*p*_interaction_ = 0.02). Community social capital can thus affect functional ability improvements variously, depending on individual psychosocial characteristics and gender. Community interventions aiming to foster social capital should focus on people who are excluded from existing opportunities to participate.

## 1. Introduction

Populations are ageing rapidly worldwide, and over a third of older people in high-income countries have a functional disability [1]. In 2016, it was reported that 27% of the population of Japan was aged 65 or over, and this is expected to reach 37% by 2050 [2]; furthermore, over six million Japanese older adults were certified as eligible for public long-term care insurance benefits because of functional disability [3]. Notably, however, a previous study conducted by the current authors found that in Japan, a fifth of older people improve their functional ability after becoming functionally disabled [4].

The World Health Organization and the Japanese government have stressed that to achieve greater improvement in functional ability, enriching local resources that facilitate older adults’ participation in communities is essential [5,6]. Specifically, enriching community social capital (i.e., “resources that are accessed by individuals as a result of their membership of a network or a group”) may play an important role in the development of suitable communities [7,8]. 

A number of studies have suggested that community social capital is associated with good perceived health [9,10], low incidence of functional disability [11], and low mortality [12,13]. However, the prospective results are still limited and are not always consistent [14]. For example, community social capital was not associated with all-cause mortality in New Zealand [15], and community social participation was not associated with change of activities of daily living (ADL) among Japanese older adults, while individual participation was associated with a change of ADL [16].

There may be cross-level interactions between community social capital and individual psychosocial characteristics. That is, the effects of community social capital may vary depending on the characteristics of subpopulations, making it beneficial for some and harmful for others. In particular, caution is needed when conducting community-empowering interventions, given the potential unfavorable side of social capital: bonding social capital can induce the exclusion of outsiders, excessive demands on community members, restrictions on individual freedoms, and downward-leveling norms [17]. Although cross-level interactions have previously been examined, these studies did not evaluate community social capital using a validated measure, mostly evaluating social capital with a single dimension and using a large geographical unit such as states in the United States or countries in Europe [18,19,20]. Moreover, these studies did not focus on the functional abilities of older people. Evaluating community social capital in small geographical units is important for the study of older people, because they may spend more time in their communities than other generations. Community social capital evaluated in small, spatial units may capture informal social relations, while that evaluated in large units, such as states and countries, may capture more political and societal structural pathways [21].

Furthermore, the association between social capital and health may differ by cultural context. Many studies so far have been conducted in Western countries, but evidence from other parts of the world, including Asia, is scarce. For example, a large-scale study in China revealed that higher community-level civic participation was associated with poor mental health in urban areas [22]. As one of the potential reasons, the authors discussed the specific Chinese culture: overwhelming pressure to engage in civic participation, especially among people with higher socioeconomic status, may lead to excess participation in social events in order to avoid a loss of “face,” which in turn increases mental stress. A study in Japan revealed that community social capital increased the incidence of functional disability, but only among women [11]. The possible mechanism is that Japanese women are more strongly connected to the residential community than men are, as they tend to spend more time in it. To the best of our knowledge, there is little research addressing the association between community social capital and improvement of functional ability in addition to cross-level interactions among older adults in Japan, the most rapidly aging country in the world.

Thus, this study examined, by measuring three components, the effect of community social capital on improvements in the functional ability of older people with disabilities, using longitudinal large-scale cohort data linked to a national long-term care system database in Japan. To perform this, we used a validated instrument with multiple indicators that was designed to measure the health-related community social capital (HR-CSC) of older adults at the school district level [23]. Specifically, we sought to examine (1) the overall contextual effect of community social capital on improvements in functional ability, and (2) the cross-level interactions between community social capital and individual psychosocial characteristics.

## 2. Materials and Methods

### 2.1. Data Sources

For this study, we used data from the Japan Gerontological Evaluation Study (JAGES) program and Japan’s Long-term Care Insurance (LTCI) database. The JAGES program was designed to investigate the social determinants of the health of older adults. The study participants were Japanese people aged 65 or older without functional impairment (which was defined as not being certified by the public LTCI system as using care services) at baseline [24]. In this study, we used data from the JAGES 2010 survey (94,358 people in 24 municipalities). We linked the JAGES 2010 survey data to the LTCI database with regard to information on levels of disability, the dates at which these levels changed, and the dates at which individuals died or moved to a different municipality. The survey period of the LTCI database ranges from 14 to 46 months, depending on municipalities. In this study, the survey population was 4234 (out of an original cohort of 94,358), as these developed a functional disability during the survey period. The follow-up period was up to 1318 days, which commenced for each person at the date they were initially certified. The exclusion criteria for the final analysis were missing data for age and gender (*n* = 85), and missing data for residence (*n* = 6). Consequently, the final analysis included 1936 men and 2207 women living in 320 communities. To assess community social capital, we used the data of 93,983 people living in 530 communities (school districts), excluding those with missing data for the area of residence (*n* = 375). The study protocol was approved by the Ethics Committee for Research on Human Subjects at Nihon Fukushi University, Japan (No. 10–05), and the Ethics Committee for Medical Research at the University of Tokyo (No. 10555).

### 2.2. Measures

#### 2.2.1. Improvement in Functional Ability

Levels of disability were objectively assessed at the time of certification for the utilization of Long-term Care (LTC) services, based on nationally standardized criteria (Appendix A) [25,26]. Each municipality’s Certification Committee for Long-term Care Needs assigned levels of disability based on both the opinion of a primary physician and a home-visit interview. Then, the levels of disability were repeatedly measured, with the second assessments being conducted within half a year, and the following assessments being conducted at least once annually, or when LTC service users or their families applied for reassessment.

There are seven levels of disability: Requiring Support-1 and -2 and Requiring LTC-l (partial support needed for basic ADLs) to LTC-5 (complete support needed for all ADLs). These measurements have frequently been used in previous studies exploring the predictors of functional disability and mortality [12,27]. The current study population only included those who were assigned to Requiring LTC-l–LTC-5 at the initial assessment, because those assigned to Requiring Support-1 and -2 are not eligible for LTCI benefits, only LTC-prevention programs. Improvement in levels of disability was defined as being determined at a follow up to have improved by one or more levels since the initial assessment. Further, transition into Requiring Support-1 or -2 was also included as an improvement in level of disability. 

#### 2.2.2. Three Components of Health-Related Community Social Capital (HR-CSC)

We assessed three components of community social capital (levels of community civic participation, social cohesion, and reciprocity) using the HR-CSC instrument [23]. Specifically, levels of community civic participation were measured by summing each individual’s participation in any of three types of community groups (volunteer groups, sport groups, and/or hobbies), where the rate of participation was once a month or more. Meanwhile, levels of community social cohesion were measured by summing the percentage of those who answered “very” or “moderately” to three items: trust (“do you think that people living in your area can be trusted, in general?”), perceptions of others’ intentions to help (“do you think that people living in your area try to help others in most situations?”), and attachment to the residential area (“how attached are you to the area in which you live?”); other answers were “neutral,” “slightly,” and “not at all.” Levels of community reciprocity were measured by summing the percentage of those who received emotional support (“is there someone who listens to your concerns and complaints?”), provided emotional support (“do you listen to others’ concerns and complaints?”), or received instrumental support (“is there someone who looks after you when you are sick and confined to bed for a few days?”). Levels of community civic participation, community social cohesion, and community reciprocity were standardized. For feasibility reasons, we used school districts as community units [23].

#### 2.2.3. Three Individual Psychosocial Characteristics

The developers of the HR-CSC defined three individual characteristics of psychosocial conditions or social relationships that are closely related to the components of community social capital: (1) participation in community groups, (2) perception of community social cohesion, and (3) social support [23]. Participation in community groups was measured by summing the number of participations in the following groups: volunteer groups, sport groups, and/or hobbies (score range: 0 to 3). Perception of community social cohesion was measured by summing the number of the following items to which the study participants answered “very” or “moderately”: trust, perceptions of others’ intention to help, and attachment to the residential area (which are the same items as those used for the measurement of community social cohesion) (range: 0 to 3). Finally, social support was measured by summing the number of the following social supports experienced: reception of emotional support, provision of emotional support, and reception of instrumental support (range: 0 to 3). Missing values were treated as 0.

#### 2.2.4. Other Covariates

Other covariates included age at initial LTC certification, income (tertiles of equivalent household income), education (≤nine or >nine years), marital status, living status, and comorbidities (history of one or more of the following diseases: stroke, heart disease, diabetes mellitus, and hypertension) [28,29]. Missing values for education, income, marital status, living status, and comorbidities were treated as dummy variables.

### 2.3. Statistical Analysis

To investigate the individual and contextual community characteristics and their cross-level interactions with respect to improvement in functional ability, we used multilevel Weibull survival models, including a community-level random intercept. We conducted analyses separately for community civic participation, social cohesion, and reciprocity. Additionally, we analyzed the data by considering its structure at two levels: the individual level (*n* = 1936 men; *n* = 2207 women) and the community level (*n* = 320). We first estimated empty models and then included the individual and community predictors (Model 1). Then, we further included cross-level individual-community interaction (Model 2). Individual psychosocial characteristics were treated as dichotomous variables (0 vs. 1 or more) because these can explicitly model meaningful conditions. We also conducted a sensitivity analysis using continuous values. To avoid multicollinearity, individual psychosocial characteristics were centered around the group (e.g., community) mean [30]. Stata 14.0 (StataCorp LP, College Station, TX, USA) was used for statistical analyses [31].

## 3. Results

The average follow-up period was 315 days (standard deviation = 269; maximum = 1318). Among the participants, 17.8% of the men and 21.1% of the women were found to have improved their functional ability at a follow-up (Table 1). For both men and women, the incidence of improvements in functional ability significantly varied across communities (empty models in Appendix A). 

Multivariate models showed that, for men, neither the main effect of community civic participation (hazard ratio (HR): 0.93, 95% confidence interval (CI): 0.78–1.12) nor the cross-level interaction effect between community civic participation and individual group participation in the community were observed with regard to improvements in functional ability (HR: 0.92, 95%CI: 0.61–1.39, *p* for interaction = 0.70) (Appendix A). For women, the cross-level interaction effect was observed (HR: 1.39, 95%CI: 0.99–1.95, *p* for interaction = 0.05), while the main effect was not (HR: 0.89, 95%CI: 0.75–1.04) (Appendix A). Among women living in communities with high civic participation, predicted mean months until improvement in functional ability was longer for those who did not participate in any civic group than for those who participated in some groups (Figure 1).

The identical models for community social cohesion indicated that, for men, the main effect of community social cohesion on improvements in functional ability was not observed (HR: 0.98, 95%CI: 0.83–1.16); however, the cross-level interaction with individual perception of community social cohesion was observed (HR: 1.71, 95%CI: 1.11–2.62, *p* for interaction = 0.02) (Appendix A). Among men living in communities with high social cohesion, predicted mean months until improvements in functional ability was longer for those whose perception of community social cohesion was lower (Figure 2).

For women, neither the main effect of community social cohesion (HR: 0.89, 95%CI: 0.75–1.06) nor the cross-level interaction effect (HR: 1.15, 95%CI: 0.81–1.63, *p* for interaction = 0.43) was observed (Appendix A).

We did not find either the main effect of community reciprocity or cross-level interactions between community reciprocity and individual social support for men (Appendix A) or women (Appendix A) (Figure 3).

A sensitivity analysis with continuous values of individual psychosocial characteristics supported these results (Appendix A).

## 4. Discussion

The results of our multilevel longitudinal study showed diverse patterns—based on individual psychosocial characteristics and gender—in the association between community social capital and improvements in functional ability. The community characteristics that facilitate civic participation may help women who participate in group activities to improve their functional ability. However, for women who do not participate in any group activities, such community characteristics may reduce the possibility of improvement in their functional ability. Similarly, in communities with high social cohesion, men who perceive their communities to be cohesive are more likely to improve their functional ability, whereas, for men who perceive their communities not to be cohesive, the cohesive community characteristics may reduce the possibility of improvement in their functional ability.

These results accord with the findings of a recent study conducted in 22 European countries, which found that when a nation has a high overall participation rate, lower self-rated health is only associated with individuals who do not participate in any types of clubs or associations [19]. Further, our study adds to the literature by showing that such cross-level interaction effects also improve the functional ability of older women. For socially active women, social participation in a variety of different types of organizations, or the presence of medical and rehabilitation services in the community, may contribute to the prevention of functional disability [32,33]. The possible positive effects of community social capital on socially active women include: (1) informal social control: residents in a cohesive community maintain social order; (2) collective efficacy: a cohesive community facilitates consensus-building efforts among community members; and (3) social contagion: health behaviors spread faster in more cohesive communities through the diffusion of information or the transmission of behavioral norms [8].

On the other hand, the potential adverse effects of community social capital on socially inactive women may reflect the drawbacks of social capital, as suggested by Portes: groups with rich, bonding social capital may exclude others outside the group [17]. Older women who do not participate in any civic group in the community may be socially excluded, which may lead to psychosocial distress, a lack of access to necessary infrastructures and services, and a lack of internal motivations to engage in functional recovery [34,35]. Among older men, such cross-level interaction was not observed. Japanese older men may be more likely than women to continue labor market participation or to participate in social groups outside their community of residence, such as alumni associations or co-worker-based activities, which is suggested by the higher college-going rate and employment rate among men than women [36].

Community social cohesion was determined to be inversely associated with improvements in the functional ability of men who did not perceive their residential communities to be cohesive. This result conforms with those of recent studies conducted in the USA and Europe, which found that in communities with high mutual trust, those who did not trust others showed lower self-rated health than those who trusted others [18,19,20]. This negative impact on individuals with low perceptions of community cohesion may also be explained by the potential drawbacks of social capital causing social exclusion in the community [17]. In highly cohesive communities, older men with lower perceptions of community social cohesion might experience social exclusion, alienation, or be ostracized by other community members [17,37]. Such psychosocial distress and lack of interpersonal interactions and relationships may hinder older men from rehabilitating their functional ability after developing a disability [35]. Among older women, such a cross-level interaction effect was not observed. Japanese older women may interact with other community members regardless of their perception of community social cohesion, because older women may spend more time in their communities than men [36,38].

Community reciprocity was determined not to be associated with improvements in functional ability, regardless of individual characteristics, including the reception/provision of social support. Almost 90% of the men and women reported that they received or provided social support, and the level of community reciprocity was measured by aggregating the responses to those questions for each community. This means that the measurement of community reciprocity applied in the current study was not suitable for detecting statistical differences in improvements in functional ability across levels of community reciprocity.

There are several limitations to the current study. First, there might be self-selection bias regarding where the respondents lived [39]. Second, improvements in functional ability may have been underestimated; there may have been a lag with regard to capturing disability improvement because the LTC service users were required to undergo clinical examinations of disability levels and to consider renewal of their levels at least once a year. Third, the LTCI database used in the current study did not include information regarding disqualification from LTCI eligibility due to the regaining of functional independence; nevertheless, the investigation showed that less than 3% had been disqualified from LTCI eligibility within four years [40]. This may have also resulted in an underestimation of functional disability improvements. Fourth, the LTCI database may not include information regarding actual changes in levels of disability after hospital admission because LTC services were not necessary during hospitalization.

## 5. Conclusions

Despite the above limitations, the current study has important implications for health policy. Community social capital can have differing impacts on the improvements in functional ability experienced by older people living in the same community, depending on individual psychosocial characteristics and gender. Community-empowering policies and actions aiming to strengthen community social capital might have negative impacts for specific vulnerable or socially isolated populations, such as those who do not participate in any group in a community. Given the findings of this study, when implementing those policies, a better understanding and pre- and post-intervention assessments involving multiple stakeholders and subpopulations are critical [41,42].

## Figures and Tables

**Figure 1 ijerph-16-01310-f001:**
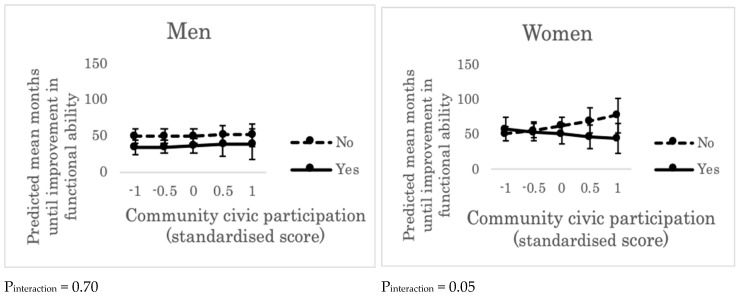
Predicted mean months until improvement in functional ability across the levels of community civic participation and individual group participation (“yes” or “no”). Error bars indicate 95% confidence intervals of predicted mean months until improvement in functional ability. P_interaction_ represents the *p* value for the cross-level interaction effect (between community civic participation and individual group participation in the community) on improvement in functional ability, adjusting for individual age, income, education, marital status, living alone, and comorbidity. “Yes”: people who participated in volunteer groups, sport groups, and/or hobbies more than once a month. “No”: people who did not participate in any such group.

**Figure 2 ijerph-16-01310-f002:**
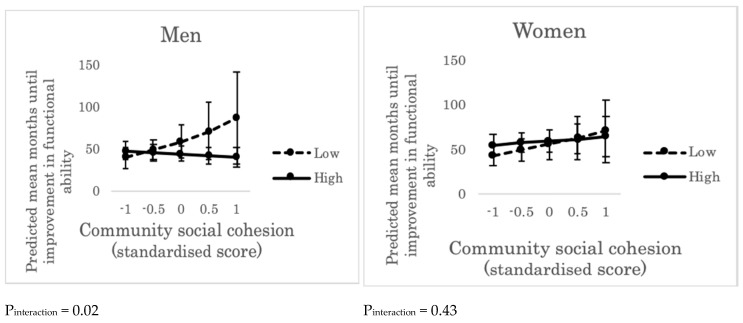
Predicted mean months until improvement in functional ability across the levels of community social cohesion and the levels of individual perception of community social cohesion (“high” or “low”). Error bars indicate 95% confidence intervals of predicted mean months until improvement in functional ability. P_interaction_ represents the *p* value of cross-level interaction effect (between community social cohesion and individual perception of community social cohesion) on improvement in functional ability, adjusting for individual age, income, education, marital status, living alone, and comorbidity. “High”: people who trusted, perceived others’ intention to help, and/or felt attached to the residential area. “Low”: people who did not trust, did not perceive others’ intention to help, and/or did not feel attached to the residential area.

**Figure 3 ijerph-16-01310-f003:**
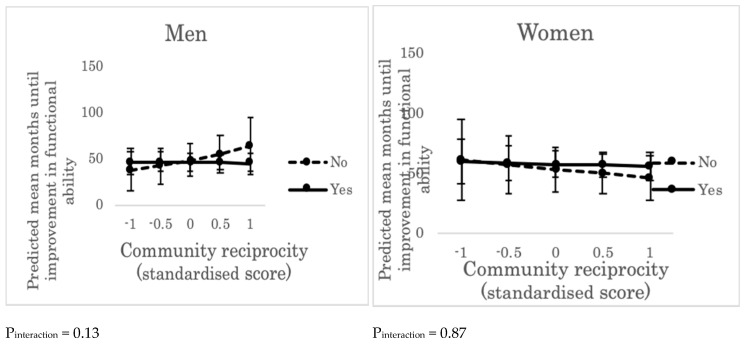
Predicted mean months until improvement in functional ability across the levels of community reciprocity and individual social support (“yes” or “no”). Error bars indicate 95% confidence intervals of predicted mean months until improvement in functional ability. P_interaction_ represents the *p* value of cross-level interaction effect (between community reciprocity and individual social support) on improvement in functional ability, adjusting for individual age, income, education, marital status, living alone, and comorbidity. “Yes”: people who received emotional support, provided emotional support, and/or received instrumental support. “No”: people who did not receive emotional support, did not provide emotional support, and/or did not receive instrumental support.

**Table 1 ijerph-16-01310-t001:** Participants’ characteristics (*n* = 4143).

Factors		Male (*n* = 1936)			Female (*n* = 2207)		
		Total	Improvement in Functional Ability		Total	Improvement in Functional Ability	
			(*n* = 345, 17.8%)			(*n* = 465, 21.1%)	
		*n*	*n*	%	*n*	*n*	%
Age							
	65–74	392	68	17.3	271	56	20.7
	75–84	1005	188	18.7	1078	237	22.0
	85 +	539	89	16.5	858	172	20.0
Income							
	T1 (lowest)	488	86	17.6	558	113	20.3
	T2	460	78	17.0	383	79	20.6
	T3 (highest)	474	84	17.7	454	100	22.0
	Missing	514	97	18.9	812	173	21.3
Education							
	0–9	973	168	17.3	1251	273	21.8
	10 +	736	136	18.5	668	135	20.2
	Missing	227	41	18.1	288	57	19.8
Marital status ^a^							
	Single/divorced/widowed	296	64	21.6	1157	255	22.0
	Married or cohabiting	1482	253	17.1	845	170	20.1
	Missing	158	28	17.7	205	40	19.5
Living alone							
	No	1617	282	17.4	1661	352	21.2
	Yes	136	27	19.9	353	76	21.5
	Missing	183	36	19.7	193	37	19.2
Comorbidity ^b^							
	No	467	84	18.0	560	135	24.1
	Yes	1148	207	18.0	1257	266	21.2
	Missing	321	54	16.8	390	64	16.4
Group participation ^c^							
	0	1578	266	16.9	1784	378	21.2
	1 +	358	79	22.1	423	87	20.6
Perception of community social cohesion ^d^							
	0	359	62	17.3	455	103	22.6
	1 +	1577	283	17.9	1752	362	20.7
Social support ^e^							
	0	210	34	16.2	231	48	20.8
	1 +	1726	311	18.0	1976	417	21.1

^a^: Single = never married. ^b^: Comorbidity = heart disease, stroke, diabetes, or hypertension. ^c^: Group participation in the community was measured by summing the number of participations in the following groups: volunteer groups, sport groups, and/or hobbies (score range: 0 to 3). ^d^: Perception of community social cohesion was measured by summing the number of following items to which the study participants answered ‘very’ or ‘moderately’: trust, perceptions of others’ intention to help, and attachment to the residential area (these were the same items as those used to measure community social cohesion) (range: 0 to 3). ^e^: Social support was measured by summing the number of the following social supports experienced by the participants: received emotional support, provided emotional support, and received instrumental support (range: 0 to 3). T: Tertile.

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
