# Peer review of "Social Capital and the Improvement in Functional Ability among Older People in Japan: A Multilevel Survival Analysis Using JAGES Data"

_ijerph, 2019, doi:10.3390/ijerph16081310_

Round 1

Reviewer 1 Report

In general, this paper is well written. The authors addressed an interesting research question. The methodology is appropriate, and the results are sound.

My major concerns are the contributions of this study. The health impact of social capital is not new in the literature. The authors must clearly indicate how this study advances our understanding.

The authors had better provide a comprehensive literature review on the nexus between social capital and health, especially the studies related to Asian countries (for example, the recently published article https://doi.org/10.3390/ijerph16040665), before presenting the methodological part.

Author Response

Thank you for your careful review of our manuscript. Please find our responses to the reviewers’ comments below and subsequent revisions in the main text in RED highlights.

We would like to thank the reviewers for their time and thoughtful comments.

Response to Reviewer 1 Comments

Point 1: In general, this paper is well written. The authors addressed an interesting research question. The methodology is appropriate, and the results are sound.

My major concerns are the contributions of this study. The health impact of social capital is not new in the literature. The authors must clearly indicate how this study advances our understanding.

The authors had better provide a comprehensive literature review on the nexus between social capital and health, especially the studies related to Asian countries (for example, the recently published article https://doi.org/10.3390/ijerph16040665), before presenting the methodological part.

Response 1:

We really appreciate this valuable comments and information. We have re-constructed a comprehensive literature review on the association between social capital and health including evidence from Asian countries, and have added descriptions about how this study contributes our understanding the impact of social capital on health as follows:

Page 2, lines 53-57 …. “However, the prospective results are still limited and are not always consistent [14]. For example, community social capital was not associated with all-cause mortality in New Zealand [15], and community social participation was not associated with change of activities of daily living (ADL) among Japanese older adults, while individual participation was associated with change of ADL [16].

Pages 2-3, lines 71-82 “Furthermore, the association between social capital and health may differ by cultural context. Many studies so far have been conducted in Western countries, but evidence from other parts of the world, including Asia, is scarce. For example, a large-scale study in China revealed that higher community-level civic participation was associated with poor mental health in urban areas [22]. As one of the potential reasons, the authors discussed the specific Chinese culture: overwhelming pressure to engage in civic participation, especially among people with higher socioeconomic status, may lead to excess participation in social events in order to avoid a loss of “face,” which in turn increases mental stress. A study in Japan revealed that community social capital increased the incidence of functional disability, but only among women [11]. The possible mechanism is that Japanese women are more strongly connected to the residential community than are men, as they tend to spend more time in it. To the best of our knowledge, there is little research addressing the association between community social capital and improvement of functional ability in addition to cross-level interactions among older adults in Japan, the most rapidly aging country in the world.

Reviewer 2 Report

Thank you for the opportunity to review the above titled manuscript. The manuscript is excellent, very well written, organized and presented and will be of great interest to others. The authors investigated the relationship of social capital with functionality among a group of senior study subjects in multiple communities of Japan. The manuscript reads very well and should be published.

First Impression:

The manuscript is very interesting, stimulating, a timely topic, informative and explored the influences of social capital on functionality of seniors with disabilities. The investigators have provided evidence of rigorous methods, significant results, informative conclusions and knowledge of the literature.  

Strengths:

The paper is very well written, organized and presented. Assertions are supported by rigorous methods and interpretation of results. Authors demonstrate a strong understanding of the literature and place their finding in appropriate context.

Weaknesses:

I find no weaknesses, this paper is very polished.

Comments and Editorial Feedback:

I offer only two comments that might improve the readability of the manuscript

Page 2, line 52, I recommend dropping in this regard

Page 2, line 54, I recommend dropping Nonetheless, … I suggest, There may be…

Author Response

Thank you for your careful review of our manuscript. We would like to thank the reviewers for their time and thoughtful comments.

Response to Reviewer 2 Comments

Comments and Editorial Feedback:

Point 1: I offer only two comments that might improve the readability of the manuscript

Page 2, line 52, I recommend dropping in this regard

Page 2, line 54, I recommend dropping Nonetheless, … I suggest, There may be…

Response:

We have amended the sentences as follows according to reviewer2’s comments.

Page 2, lines 49-51, “Specifically, enriching community social capital (i.e., ‘resources that are accessed by individuals as a result of their membership of a network or a group’) may play an important role in the development of suitable communities.”

Page 2, lines 58-59, “There may be cross-level interactions between community social capital and individual psychosocial characteristics.”

Round 2

Reviewer 1 Report

The authors have made significant improvements. I am satisfied with the quality of this revised version. I would recommend it for publication.